# Exploring the Relationship between (De-)Centralized Health Care Delivery, Client-Centeredness, and Health Outcomes—Results of a Retrospective, Single-Center Study of Transgender People Undergoing Vaginoplasty

**DOI:** 10.3390/healthcare11121746

**Published:** 2023-06-14

**Authors:** Andreas Koehler, Bernhard Strauss, Peer Briken, Margit Fisch, Armin Soave, Silke Riechardt, Timo O. Nieder

**Affiliations:** 1Institute for Sex Research, Sexual Medicine and Forensic Psychiatry, University Medical Center Hamburg-Eppendorf, Martinistrasse 52, 20246 Hamburg, Germany; 2Department for Urology, University Medical Center Hamburg-Eppendorf, Martinistrasse 52, 20246 Hamburg, Germany; 3Institute of Psychosocial Medicine, Psychotherapy and Psycho-Oncology, University Hospital Jena, Stoystrasse 3, 07740 Jena, Germany; 4Interdisciplinary Transgender Health Care Center, University Medical Center Hamburg-Eppendorf, Martinistrasse 52, 20246 Hamburg, Germany

**Keywords:** transgender health, health care delivery, gender-affirming surgery, vaginoplasty, health care research

## Abstract

*Introduction*: Transgender health care interventions (e.g., gender-affirming surgery) support transgender and gender-diverse people to transition to their gender and are delivered in both centralized (by one interdisciplinary institution) and decentralized settings (by different institutions spread over several locations). In this exploratory study, we investigated the relationship between centralized and decentralized delivery of transgender health care, client-centeredness, and psychosocial outcomes. *Methods*: A retrospective analysis of 45 clients undergoing vaginoplasty at one medical center was conducted. Mann–Whitney U tests assessed differences regarding five dimensions of client-centeredness and psychosocial outcomes between the health care delivery groups. To address shortcomings regarding the small sample size, we applied a rigorous statistical approach (e.g., Bonferroni correction) to ensure that we only identified predictors that were actually related to the outcomes. *Results*: All aspects of client-centered care were scored average or high. Decentralized delivery of care was more client-centered in terms of involvement in care/shared decision-making and empowerment. However, participants from decentralized health care delivery settings scored lower on psychosocial health (*p* = 0.038–0.005). *Conclusions*: The factor of (de-)centralized health care delivery appears to have a significant impact on the provision of transgender health care and should be investigated by future research.

## 1. Introduction

Transgender and gender-diverse (TDG) people experience their gender identity as incongruent to their sex assigned at birth (ICD-11: Gender Incongruence), which can be associated with clinically relevant distress (DSM-5-TR: Gender Dysphoria). Around 80% of TGD people identify as the binary ‘opposite’ gender (male or female), whereas the other 20% identify as non-binary (e.g., genderfluid) [1,2,3].

Gender dysphoria can lead a person to undergo certain medical procedures to alter their primary and secondary sex characteristics and live according to their gender identity. Therefore, a variety of effective transition-related medical interventions are offered by providers of transgender health care [4,5]. Besides counselling to accompany transitioning and mental health care for concomitant mental health problems, hormonal therapy and gender-affirming genital surgery are common transition-related interventions. Breast removal or augmentation, hair removal, speech therapy, phonosurgery, and facial surgery are further interventions in support of the transition process [4,5]. All these interventions are considered essential and effective options for TGD people to reduce gender dysphoria and have been found to improve mental health and quality of life [6,7,8,9,10,11,12].

Currently, transgender health care is delivered in both centralized and decentralized settings. Specialized interdisciplinary centers providing all transition-related interventions at one location are considered centralized settings of transgender health care delivery. These institutions have been found to be more common in European countries [13]. Transgender health care is delivered in a decentralized manner when transition-related interventions are provided by different institutions, localized at various places [14,15,16]. Historically, transgender health care delivery has evolved differently in Europe and North America, whereas in other countries, especially those of the global south, structures are not even sufficiently developed to ensure a certain degree of access to specific care for TGD people [17]. A more detailed discussion of the issue can be found in another publication by the authors of the present study [13]. It has been found that both centralized and decentralized delivery of transgender health care can have certain distinct advantages (e.g., regarding the training of health care providers) and disadvantages (e.g., regarding accessibility) and might affect both the quality and the provision of client-centered transition-related health care [13,17].

Client-centered health care is conceptualized as putting the specific health needs of a person seeking treatment and their desired health outcomes at the center of medical decision-making and quality measurements [18]. When decisions need to be made where several equivalent treatment options are available (so-called equipoise), individual health needs and related treatment preferences are recommended to be preferred over criteria-based checklists [19]. In line with this, clients are considered partners with their health care providers and are not only treated from a clinical perspective. Instead, providers consider a client’s emotional, mental, social, and financial perspectives, and also consider additional aspects that might be relevant (e.g., a clients’ spiritual perspectives) [18]. The integrative model of client-centeredness summarizes five central factors of client-centered health: client as a unique person, client involvement in care, client information, clinician-client communication, and client empowerment [20,21].

In an expert survey conducted by the authors of this paper, the centralized delivery of transgender health care was expected to ensure comprehensive interdisciplinary care delivered by providers with vast professional experience. On the other hand, the decentralized delivery of transgender health care was assessed to be more suitable to address the individual needs of clients, e.g., due to better opportunities to choose certain providers. Prior research on the influences of centralized and decentralized delivery of care has mostly been conducted to investigate the issue from the perspective of health care policy, e.g., regarding the financial benefits of decentralizing care [22]. In daily medical practice, decentralized delivery of care has been found to be suitable for certain conditions (e.g., HIV), as it ensures easy access to care [23,24]. However, regarding transgender health care, centralized and decentralized delivery of care has not yet been further investigated. As the setting of transgender health care delivery differs significantly between and within different countries, it could be considered a potential influence on the outcome and quality of care [13].

There is a growing body of research investigating client-centeredness in the provision of transition-related interventions [25,26,27,28,29]. However, approaches based on specific models and studies investigating the potential intersections with systemic factors (e.g., centralized and decentralized health care delivery) are still rare. To gain a more comprehensive understanding of client-centered transgender health care, these issues need to be addressed in health services research [13,30,31].

Thus, within the present study, we explored the relationship between centralized and decentralized delivery and the client-centeredness of transgender health care. Using the integrative model of client-centeredness [20,21], we aim to contribute to a nuanced understanding of client-centered transgender health care and its association with health outcomes of transition-related medical interventions.

## 2. Methods

### 2.1. Study Design

The present retrospective study was conducted by the Institute for Sex Research, Sexual Medicine and Forensic Psychiatry, and the Department for Urology, both part of the Interdisciplinary Transgender Health Care Center at the University Medical Center hamburg-Eppendorf, Germany. It was performed according to a single surgeon’s experience (S.R.). The STROBE statement can be found in the Appendix A. The study was part of a larger research project on client-centered transgender health care [31]. The study received ethical approval by the Chamber of Psychotherapists Hamburg Ethics Committee (10/2018-PTK-HH).

#### 2.1.1. Participants

Participants needed to be at least 16 years of age and have undergone a two-step vaginoplasty using penile inversion technique to be eligible for study participation. All former clients who underwent vaginoplasty between 2013 and 2018 were invited to participate. Written consent was obtained from all participants.

#### 2.1.2. Participant Recruitment

Data were collected between January and March 2020. We identified all eligible former clients and invited them to participate in the study by letter. The invitation letter contained information on the study and the weblink to the online survey. If participants did not want to fill out the survey online, we offered the opportunity to participate by mail or to answer the survey on a desktop computer at the medical center. Participants needed to give their informed consent before answering the survey. Of the 119 eligible former clients, 3 could not be traced. 116 former clients were contacted and asked to participate in the study. A total of 45 former clients were included in the study (response rate of 38.8%). All participants answered at least 90% of the survey questions. We were not able to determine why former clients who declined to participate did not want to participate in the study. However, a non-responder analysis was performed to assess a systematic bias in the recruitment procedure. We compared data on age and population of place of residence between participants and non-participants. We determined the population of the place of residence of non-participants by analyzing their postal address, whereas participants answered a question concerning this issue. Due to the method of participant recruitment, access to a web-enabled device and technical affinity need to be considered as potential biases.

### 2.2. Measures

The study assessed sociodemographic data, data regarding undergone and planned treatments, gender congruence, mental health outcomes, quality of life, clinical surgery outcomes, and outcomes concerning dimensions of client-centered care. Participants were considered to have received care in a centralized health care delivery setting when they received counselling and other treatments within the Interdisciplinary Transgender Health Care Center. Those who only underwent vaginoplasty at the Department for Urology were considered to have accessed decentralized transgender health care [31]. The present analysis investigated five central aspects of client-centered care [21]: client as a unique person (Trust in Physician Scale, TiPS [32,33]), client involvement in care (Shared Decision-Making Questionnaire, SDM-Q-9 [34]), client information (self-constructed questionnaire), clinician-client communication (Quality of Physician-Patient Interaction, QQPPI [35]), and client empowerment (Health Care Empowerment Inventory, HCEI [36]). The 11 items of the TiPS were scored on a 5-point Likert scale ranging from 1 (strongly disagree) to 5 (strongly agree). The sum score was obtained by taking the unweighted mean of the responses to the items and transforming that value to a 0–100 scale. The 9 items of the SDM-Q-9 were scored on a 6-point Likert scale ranging from 1 (strongly disagree) to 6 (strongly agree). The sum score was ranged on a scale from 0 to 100. Items of the questionnaire on patient information were scored on a Likert scale ranging from 1 (strongly disagree) to 5 (strongly agree). We calculated an unweighted mean score over all items. The same was true for the QQPPI. Items of the HCEI were scored on a Likert scale ranging from 1 (strongly disagree) to 5 (strongly agree). We calculated an overall sum score for this questionnaire. We assessed the relation of client-centered care to basic demographic aspects, gender congruence (Transgender Congruence Scale, TCS [37]), quality of life (WHOQOL-BREF [38]), and psychological distress (BSI-18 [39]). The TCS items were scored on a 5-point scale. We calculated an unweighted mean score using all items. The WHOQOL-BREF measures quality of life according to four dimensions (physical health, psychological health, social relationships, environment) for 26 items. The items are scored on a 5-point Likert scale. The scores were transformed and scaled from 0 to 100. The 18 items of the BSI-18 were rated on a 5-point Likert scale. The items were summarized to the Global Severity Index (GSI). All measures were chosen based on systematic reviews of the psychometric qualities of the questionnaires and the clinical experience of the research group.

#### Data Analysis

All statistical analyses were conducted using SPSS 28. Missing data were deleted pairwise. Sample characteristics were reported descriptively. We used Shapiro–Wilk tests to check for normal distribution of our outcome variables. Mann–Whitney U tests were performed to assess the differences regarding the five dimensions of client-centeredness, gender congruence, quality of life, and psychological distress between the health care delivery groups (centralized vs. decentralized). We used G*Power 3.1.9.7 to determine the sample size necessary to find differences between the two groups using a Mann–Whitney U test for an effect size of 0.80 and a power of 0.80. We calculated η^2^ to assess the effect size. An effect of 0.01 was considered small, 0.06 was considered medium, and 0.14 or higher was considered large [40]. For the non-responder analysis, a *t* test for independent samples and a chi-square test was performed. We compared age and population of place of residence between participants and non-participants. All analyses were performed with an alpha level of 0.05, and—to deal with the problem of multiple comparisons—a Bonferroni-corrected [41] alpha level of 0.005 (0.05 divided by the 11 group comparisons calculated [42]).

## 3. Results

The demographic characteristics are presented in Table 1. Most participants were born in Germany, lived in places with more than 1,000,000 residents, were single, highly educated, and full-time employed.

Table 2 provides information on the participants’ genders. Most participants identified as women or transwomen. Three participants reported a non-binary gender. Table 3 summarizes undergone and planned transition-related interventions. All participants underwent mental health counseling, hormone treatment, and feminizing genital surgery (vaginoplasty).

“Client involvement in care” was scored average. All other aspects of client-centered care were rated high (Table 4). A Shapiro–Wilk test indicated that our outcome variables did not follow normal distribution, W(45) = 0.797–0.946, *p* = 0.035–0.000. The sample size for two groups (centralized and decentralized delivery of health care) necessary to find differences using Mann–Whitney U tests for an effect size of 0.80 and a power of 0.80 was 21 for each group. Our groups met these requirements. Differences were found between the group accessing transgender health care in a centralized delivery setting and the decentralized group for the subdimensions “client involvement in care/shared decision-making” and “empowerment”. Moreover, the decentralized group reported lower scores on the “physical health” dimension of quality of life and higher psychological distress (Table 4). All differences were statistically significant on an alpha level of 0.05 but did not survive Bonferroni correction.

The non-responder analysis revealed no significant differences between participants and non-participants with regard to age (*t*(115) = 0.166, *p* = 0.868) and the population of the place of residence (χ^2^ (4, N = 116) = 0.810, *p* = 0.937).

## 4. Discussion

As far as we know, this is the first study that investigated the relationship between centralized and decentralized delivery of transgender health care and client-centeredness of transition-related interventions.

The present sample was comparable to those examined in prior research with regard to various demographical variables, e.g., age, education [1,43,44]. It could have been assumed that participants living in places with a smaller population were more likely to access transgender health care in a decentralized setting. Due to the small sample size, we were not able to analyze this question statistically. However, a univariate analysis of the data—published in another journal [45]—does not support that assumption. Participants from both centralized and decentralized health care delivery settings were living in rural and urban areas in comparable numbers. We found that our sample was also comparable to other samples of cisgender people accessing medical care regarding the evaluation of the different dimensions of client-centered health care [33,36,46]. Our sample scored even higher regarding clinician–client communication [35,47]. As gender-affirming genital surgeries are intense and complex interventions, a comprehensive and detailed communication between providers and clients seems to be of particular importance, which could be why our sample scored higher on this dimension. We also found differences in the assessment of client-centeredness between participants accessing transition-related interventions in centralized and decentralized delivery settings.

Participants who underwent a vaginoplasty in a decentralized health care delivery setting reported stronger involvement in care and felt more empowered compared to those from a centralized delivery setting. To be involved in care means that there is an expectation for clients to actively participate in the decision-making process, and to share information and personal values. In the end, the client and the provider achieve a tailored treatment decision with shared responsibility [48]. The performing surgeon or other health care professionals might have given clients from the decentralized setting more comprehensive preoperative client information about the intervention and the center, as they had not accessed medical care at that particular institution beforehand. Moreover, clients from the decentralized delivery setting could also have demanded more information from the professionals about the intervention and the institution. Consequently, they experienced a higher degree of involvement in the decision-making and felt more empowered. Prior research expected the decentralized delivery of transgender health care to be more individual, e.g., because it gives the client several opportunities to choose their health care provider [13]. However, this approach might also require greater personal responsibility when making decisions about one’s own health care, which might be why these individuals reported higher scores for the empowerment dimension. Nevertheless, even though participants from the decentralized setting reported health care as more client-centered, they also reported higher amounts of psychological distress and a lower quality of life regarding physical health. In an expert survey conducted by the authors of this paper, the potential disadvantages of the decentralized delivery of transgender health care were found to be a potential lack of expertise and training of health care professionals and the fragmentation of care. Therefore, postsurgical problems of participants from the decentralized group that could not be handled immediately by the surgeon, e.g., psychological distress caused by unsatisfying aesthetical results, might not have been addressed properly by their other health care providers. In centralized settings, health care professionals with a higher amount of clinical expertise might have been more aware of the specific challenges that might arise after gender-affirming surgery [13]. However, it is important to note that health care providers in decentralized delivery settings do not necessarily have inferior training or knowledge or act in a less patient-centered manner. Additionally, health care providers with extensive training do not necessarily act in a more patient-centered manner or respect TGD peoples’ individual needs. Therefore, this relationship needs to be explored more deeply in future studies. As specialized centers offering centralized health care are mostly located in metropolitan areas [13], TDG people from rural areas might also have no alternative to accessing transition-related interventions in decentralized settings. That TGD people from rural areas tend to report higher levels of mental health problems [49] might be an additional factor that influenced the present result. Unfortunately, there is no comparable evidence from studies with cisgender populations, as the (de-)centralized delivery of health care has only been researched from a systemic perspective thus far (e.g., regarding cost effectiveness), but not in terms of client-centeredness. Future research should also take this gap in knowledge into account.

This study investigated the relationship between the centralized and decentralized delivery of transgender health care and client-centeredness. We introduced the factor of (de-)centralized health care delivery into empirical transgender health care research for the first time and found that it might play a meaningful role in the quality of transgender health care, especially regarding the client-centeredness of transition-related interventions. Our study should be understood as an exploratory attempt to shed light on the role of the health care delivery setting in regard to client-centered transgender health care. It should encourage future research to investigate these factors in more detail.

## 5. Limitations of This Study

Even though we found differences in both client-centeredness and psychosocial outcomes between our groups, most of these differences did not survive Bonferroni correction. Bonferroni correction is an established, rigorous statistical approach to reduce the risk of type I errors (i.e., to ensure the identification of only the predictors that are actually related to the outcomes). However, it is very conservative and causes power (the proportion of the false null hypothesis that is rejected correctly) to be reduced. The failure to reject the null hypothesis when it is false can lead to overly conservative conclusions drawn from research results. Therefore, our results also need to be discussed in the light of the everyday clinical experience of health care providers and need to be replicated in studies with higher statistical power (e.g., by increasing the sample size). The number of participants in our study was small, which is why the Mann–Whitney U tests we calculated were only able to find large effects with a sufficient power. Additionally, we investigated TGD people who underwent vaginoplasty only. Vaginoplasty is a medically complex transition-related genital surgery [5] that is oftentimes undergone after other interventions have already been accessed (e.g., hormone treatment). Moreover, the procedure itself might have an influence on how patient-centeredness is perceived. Therefore, the protocol should be transferred to research other gender-affirming procedures (e.g., phalloplasty) and be adapted to prospective designs. Additionally, there are also different experiences and considerations for individuals undergoing vaginoplasty depending on their age (e.g., for how long someone experienced gender dysphoria, surgical considerations, etc.) that might have influenced the outcomes of the present study. However, another analysis of the data focusing on the surgical outcome did not find the ages to be different between participants accessing vaginoplasty in centralized and decentralized settings [45]. In addition, post-surgical complications might have influenced the assessment of patient-centeredness as well. A recent analysis by authors of the present study found that even though there is a considerable rate of smaller short-term post-surgical complications (41%), the number of severe complications is low (2% [50]). Systematic reviews of the current evidence found a comparable number [51,52]. Additionally, a recent systematic review found that a majority of over 90% of TGD people undergoing vaginoplasty report high satisfaction with the procedure and a regret rate of only 2% [51]. However, it is unclear if the high satisfaction with the procedure can be directly transferred to a high satisfaction with the various aspects of health care quality (e.g., client-centeredness). Additionally, there is high subjectivity regarding the assessment of regret. It has been found that the most common reasons to regret undergoing gender-affirming surgery were dissatisfaction with the surgical result and difficulties in life with the new gender role [53]. However, even though it appears that a small percentages of TGD individuals experience regret after gender-affirming surgery for different reasons, the majority experiences a significant alleviation of gender dysphoria and increased self-acceptance, and therefore do not regret undergoing those procedures. To reach a better understanding of the reasons that TGD people might regret undergoing gender-affirming procedures, standardized, validated questionnaires should be developed. Multi-center studies might be necessary to deal with the potential effects of a clinic’s individual surgeon(s). All of the participants of the current study underwent vaginoplasty be a single surgeon. Therefore, potential selection effects (e.g., specific patient population) and confounding variables (e.g., surgeon’s experience, surgical approach) cannot be ruled out. In particular, a surgeon’s experience is oftentimes assumed to be associated with the surgical outcome. However, even though there is no data published investigating this question for gender-affirming surgery, the evidence from other surgical procedures is mixed [54,55] and differences were often explained by surrounding factors (e.g., patient’s emergency status [55]). Whether these findings can be transferred to gender-affirming procedures must be investigated by future research. As the data were generated in Germany, a country that is ethnically rather homogenous, greater racial and ethnic diversity could also alter the results significantly. Moreover, Germany’s system of mandatory health insurance founded by general wage contributions might influence the data compared to countries such as the United States, with a mix of public and private for-profit and non-profit insurers.

## Figures and Tables

**Table 1 healthcare-11-01746-t001:** Demographic characteristics.

	No., %
**N**	45 (100.0)
**Age, Mean (SD)**	43.4 (15.6)
**Country of birth**	
Germany	38 (84.4)
Other European countries	2 (4.4)
Non-European countries	2 (4.4)
Cannot or do not wish to answer this question	3 (6.6)
**Population of the place of residence**	
<5000	5 (11.1)
5000–20,000	6 (13.3)
20,000–100,000	6 (13.3)
100,000–1,000,000	3 (6.6)
>1,000,000	17 (37.8)
I do not know	3 (6.6)
Cannot or do not wish to answer this question	5 (11.1)
**Marital status**	
Single	19 (42.2)
in a relationship	9 (20.0)
married, living together	5 (11.1)
married, living separately	2 (4.4)
registered partnership, living together	2 (4.4)
Divorced	3 (6.6)
Widowed	2 (4.4)
Cannot or do not wish to answer this question	3 (6.6)
**Education**	
Low	8 (17.8)
Middle	12 (26.7)
High	20 (44.4)
Cannot or do not wish to answer this question	5 (11.1)
**Employment**	
Full-time	17 (37.8)
Part-time	5 (11.1)
Mini job (i.e., individual earnings < 400 €/month)	4 (8.8)
Unemployed	6 (13.3)
Retired	4 (8.8)
Cannot or do not wish to answer this question	9 (20.0)
**Occupational status**	
Student	2 (4.4)
Vocational training	4 (8.8)
Unskilled worker	2 (4.4)
Operative	1 (2.2)
Employee	22 (48.9)
Civil servant	1 (2.2)
Self-employed	3 (6.6)
Cannot or do not wish to answer this question	10 (22.2)

**Table 2 healthcare-11-01746-t002:** Gender- and treatment-related characteristics.

Gender Identity	N (%)	
Woman/female	39 (88.6)	
Transwoman	14 (31.8)	
Trans	4 (9.1)	
Transgender	6 (13.6)	
Transsexual	5 (11.4)	
Genderfluid	1 (2.3)	
Other	Androgynous	1 (2.3)
	Woman with transsexual background	1 (2.3)
	Non-binary/enby, femby, demiflux	1 (2.3)
	Transident	1 (2.3)
Non-binary gender		
No	41 (93.2)	
Yes	3 (6.8)	

**Table 3 healthcare-11-01746-t003:** Undergone and planned transition-related interventions.

Treatment	Undergone, N (%)	Planned, N (%)
Mental health counselling	45 (100.0)	6 (9.8)
Hormone treatment	45 (100.0)	45 (100.0)
Hair removal	29 (65.9)	19 (43.9)
Speech therapy	26 (56.1)	7 (17.1)
Top surgery	18 (39.0)	8 (19.5)
Feminizing genital surgery	45 (100.0)	1 (2.4)
Adam’s apple surgery	4 (9.8)	4 (12.2)
Phonosurgery	1 (2.4)	9 (19.5)
Facial surgery	1 (2.4)	9 (22.0)
Hair transplant	1 (2.4)	2 (4.9)
Others (one answer each)	0 (0.0)	0 (0.0)
**Treatment progress (ITPS *)**		
Mean (SD)	0.78 (0.18)	
Range	0.38–1.00	

* The ITPS (Individual Treatment Progress Score) ranges between 0 and 1. A higher score indicates a more advanced treatment [1].

**Table 4 healthcare-11-01746-t004:** Health care delivery setting, client-centered care, and health outcomes.

	Total Sample (Mdn)	Centralized Health Care Delivery, (Mdn)	Decentralized Health Care Delivery, (Mdn)	Statistics
**N**	45	24	21	
**Dimensions of client-centered care**				
client as a unique person	87.27	85.45 (mean rank = 21.29)	89.09 (mean rank = 24.95)	U = 211.000; *p* = 0.350, η^2^ = 0.020
client involvement in care/shared decision-making	60.0	51.11 (mean rank = 19.15)	62.22 (mean rank = 27.40)	U = 159.500; *p* = 0.035, η^2^ = 0.101
client information	4.64	4.63 (mean rank = 23.19)	4.73 (mean rank = 22.79)	U = 256.500; *p* = 0.916, η^2^ = 0.000
Clinician–client communication	4.36	4.21 (mean rank = 20.94)	4.57 (mean rank = 25.36)	U = 202.500; *p* = 0.259, η^2^ = 0.030
client empowerment	32.00	30.50 (mean rank = 17.85)	33.00 (mean rank = 28.88)	U = 128.500; *p* = 0.005, η^2^ = 0.181
**Health outcomes**				
Gender congruence [37]	4.50	4.50 (mean rank = 22.60)	4.50 (mean rank = 23.45)	U= 242.500; *p* = 0.826, η^2^ = 0.001
Quality of life (physical health) [38]	13.14	13.71 (mean rank = 26.33)	12.57 (mean rank = 18.31)	U = 329.500; *p* = 0.038, η^2^ = 0.098
Quality of life (psychological) [38]	15.27	15.33 (mean rank = 25.26)	14.67 (mean rank = 19.48)	U = 305.000; *p* = 0.134, η^2^ = 0.051
Quality of life (social relationships) [38]	14.67	14.67 (mean rank = 22.24)	16.00 (mean rank = 22.79)	U = 235.500; *p* = 0.887, η^2^ = 0.000
Quality of life (environment) [38]	16.00	17.00 (mean rank = 24.50)	15.50 (mean rank = 20.31)	U = 287.500; *p* = 0.279, η^2^ = 0.027
Psychological distress [39]	2.00	1.00 (mean rank = 18.52)	4.00 (mean rank = 26.86)	U = 150.000; *p* = 0.029, η^2^ = 0.108

## Data Availability

We will consider sharing de-identified, individual participant-level data that underlie the results reported in this article on receipt of a request detailing the study hypothesis and statistical analysis plan. All requests should be sent to the corresponding author. The corresponding author and lead investigators of this study will discuss all requests and make decisions about whether data sharing is appropriate based on the scientific rigor of the proposal. All applicants will be asked to sign a data access agreement.

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
