# Peer review of "Exploring the Relationship between (De-)Centralized Health Care Delivery, Client-Centeredness, and Health Outcomes—Results of a Retrospective, Single-Center Study of Transgender People Undergoing Vaginoplasty"

_healthcare, 2023, doi:10.3390/healthcare11121746_

Round 1
Reviewer 1 Report
Thank you for the opportunity to review the manuscript entitled, “Exploring the relationship between (de)centralized health care 2 delivery, client-centeredness, and health outcomes – Results of 3
a retrospective, single-center study of transgender people undergoing vaginoplasty” for possible publication in Healthcare. This manuscript reports on retrospective health record data from transgender women undergoing vaginoplasty in either centralized or decentralized care contexts. The associations between patient centered care, decentralized or centralized care and health outcomes were tested. This manuscript adds important and valuable information to the understanding of care contexts for transgender women seeking complex and intense gender affirming surgical interventions. However, there are several minor and major concerns to be addressed before this manuscript can be considered for possible future publication. These concerns are listed below.
Minor Concerns:
Page 2, lines 67-68: “This is especially…” This sentence has a word missing or a grammatical issue. Please revise for clarity.
Page 2, lines 71: “…providers take over the client’s emotional….” Please revise for clarity and accuracy.
Page 2, line 75: New topics being introduced; please start a new paragraph.
Page 2: line 81: Please revise introduction to make clear the theory that is driving this study. As currently written, no theory is presented, yet authors indicate that theory-driven research is rare and much needed in this area. If integrative model of client centeredness is the theory driving the research, please more clearly articulate that it is a theory. Currently the manuscript background reads as though the authors mean to communicate that client centeredness is an important strategy or approach to providing care, not necessarily a theory.
Page 3, line 126: Unclear why authors use the word, “maybe…”. Please remove and revise for clarity.
Page 7, line 184: Table 3 is referenced incorrectly twice in this paragraph. Please revise to refer to table 4.
Major Concerns:
The background communicates that this manuscript will be about the role of decentralized or centralized care for transgender people (generally). But in fact, this manuscript is much more specific as it focuses on the role of these care features for transgender women undergoing a very extensive gender affirming surgery. It would be stronger if the entire introduction was focused on transgender women’s care, rather than broadly on transgender care. This is suggested because centralized and decentralized care and patient centered care may be experienced very differently when patients receive different forms of gender affirming care. For example, it is conceivable that these features may have different associations with outcome variables for care that is not surgical compared to surgical care.
The authors seem keen to communicate that understanding the role of centralized vs decentralized care is an important aspect of gender affirming care provided to gender diverse people, and in particular, transgender women. However, the entire introduction is spent on describing and legitimizing the existence of transgender people, rather than on introducing the scientific importance of understanding the role of centralized vs decentralized sources of patient centered care and in particular the importance to transgender women seeking vaginoplasty; an intense and complex gender affirming intervention. This needs to be more clearly parsed out and introduced to the reader.
Please edit as requested in feedback above.
Author Response
Page 2, lines 67-68: “This is especially…” This sentence has a word missing or a grammatical issue. Please revise for clarity.
RESPONSE: Thank you for your remark. We found that the sentence does not add any valuable information and that it might have been hard to read. Therefore, we deleted it.
Page 2, lines 71: “…providers take over the client’s emotional….” Please revise for clarity and accuracy.
RESPONSE: Thank you for your remark. We changed the sentence as follows:
“Instead, providers take over a clients emotional, mental, social, and financial perspective, and consider additional aspects that might be relevant (e.g., spiritual perspectives).”
Page 2, line 75: New topics being introduced; please start a new paragraph.
RESPONSE: Thank you for the remark. Changed.
Page 2: line 81: Please revise introduction to make clear the theory that is driving this study. As currently written, no theory is presented, yet authors indicate that theory-driven research is rare and much needed in this area. If integrative model of client centeredness is the theory driving the research, please more clearly articulate that it is a theory. Currently the manuscript background reads as though the authors mean to communicate that client centeredness is an important strategy or approach to providing care, not necessarily a theory.
RESPONSE: Thank you for the remark. You are correct that the term theory might have been misleading. We intended to point out that prior research was not following specific models about client-centeredness, but only focused on single aspects of the concept. In contrast, our study tried to apply a more comprehensive approach by using the integrative model of client-centeredness to research the concept more in line with its complexity. Accordingly, we changed the terminology from “theory-driven” to “approaches based on comprehensive models”:
“Even though there is a growing body of research investigating client-centeredness in the provision of transition-related interventions,1-5 approaches based on comprehensive models and studies investigating the potential intersections with systemic factors (e.g., centralized, and decentralized health care delivery) are still rare.”
Page 3, line 126: Unclear why authors use the word, “maybe…”. Please remove and revise for clarity.
RESPONSE: Thank you for the remark. You are correct as the sentence was misleading. All participants received other treatments. We deleted “maybe”.
Page 7, line 184: Table 3 is referenced incorrectly twice in this paragraph. Please revise to refer to table 4.
RESPONSE: Thank you for the remark. Please excuse that mistake. We changed the references to the table to “table 4”.
Major Concerns:
The background communicates that this manuscript will be about the role of decentralized or centralized care for transgender people (generally). But in fact, this manuscript is much more specific as it focuses on the role of these care features for transgender women undergoing a very extensive gender affirming surgery. It would be stronger if the entire introduction was focused on transgender women’s care, rather than broadly on transgender care. This is suggested because centralized and decentralized care and patient centered care may be experienced very differently when patients receive different forms of gender affirming care. For example, it is conceivable that these features may have different associations with outcome variables for care that is not surgical compared to surgical care.
RESPONSE: Thank you for this important remark. You are absolutely correct that centralized and decentralized care might be experienced differently depending on the interventions undergone. However, we are unsure if a reader not familiar with transgender healthcare is aware of all the possible treatment options and if they might miss some context information if we solely focus on feminizing surgical care. Therefore, we would favor to stick to the broader perspective in the introduction. However, we added a section to the limitations, discussing the issue you raised. We added the following section:
“Also, we investigated TGD people who underwent vaginoplasty only. Vaginoplasty is a medically complex transition-related genital surgery6 that is oftentimes undergone after other interventions have already been accessed by a person before (e.g., hormone treatment). Moreover, the procedure itself might have an influence on how patient-centeredness is perceived. Therefore, the protocol should be transferred to research other gender-affirming procedures (e.g., phalloplasty) and be adapted to prospective designs.”
The authors seem keen to communicate that understanding the role of centralized vs decentralized care is an important aspect of gender affirming care provided to gender diverse people, and in particular, transgender women. However, the entire introduction is spent on describing and legitimizing the existence of transgender people, rather than on introducing the scientific importance of understanding the role of centralized vs decentralized sources of patient centered care and in particular the importance to transgender women seeking vaginoplasty; an intense and complex gender affirming intervention. This needs to be more clearly parsed out and introduced to the reader.
RESPONSE: Thank you for this remark. As we do not expect every reader to know about the specifics of transgender healthcare and transgender and gender-diverse populations in general, we aim to give a comprehensive overview over the existing evidence on the diversity of the population, the manifold healthcare options, and how these transition-related interventions can improve transgender people’s lives. Regarding the role of centralized and decentralized care, we point out in the introduction that there is only very limited research on the issue. Therefore, our study aims to add evidence to close that gap in research and clarify if and how centralized and decentralized care affects the quality of care. To make this clear, we added the following section:
“Prior research on the influence of centralized and decentralized delivery of care has mostly been conducted investigating the issue from the perspective of healthcare policy, e.g., regarding financial benefits of decentralizing care7. In daily medical practice, decentralized delivery of care has been found to be suitable for certain condition (e.g., HIV), as it ensures easy access to care8,9. However, regarding transgender healthcare, centralized and decentralized delivery of has not been further investigated yet. As the setting of transgender healthcare delivery differs significantly between and within different countries, it could be considered a potential influence on the outcome and quality of care10.”
Reviewer 2 Report
This is very timely and important contribution to the literature. I appreciate the emphasis on how we can help more individuals access gender affirming care (GAC). It is also good to know and continue to document that all patients who received vaginoplasty in the study were largely pleased with results.
Specific notes
Introduction
· Line 49: there’s a formatting issue here – the font changed
· Paragraph that beings on line 52: I would like to understand more about the socioeconomic and cultural factors influencing the centralized v. uncentralized. Is it the richer countries that are more likely to offer GAC in large academic medical institutions? What is the history of this discrepancy? As we consider the implications for folks from the global south, these sociopolitical factors are important. “Common in European countries” has a lot under the surface
· The introduction would benefit from improved transitions between sections
· This is a good introduction to client centered care, more information about how transgender patients have experienced care historically and contextually would underscore the need for this approach
· Please provide more context for why each measure was chosen
Method
· Please speak to the impact of having only single surgeon’s data set
· Please speak to the impact of having a vaginoplasty at 16 v 40 developmentally.
· The statistical analysis of this paper is beyond my scope of knowledge to opine on specifically.
Results
· Given that the sample is 85% German, it may behoove the generalization of the paper to focus on the population at hand.
· Can you please say more about the separation between woman/female and trans woman?
Discussion
· Decentralized settings seem to offer a lot, though as the discussion points out, wrap around care may offer more in terms of psychological and general health support. Is higher personal responsibility positive?
· Post surgical problems are a huge factor in future satisfaction of care – can you please speak more to this?
· The discussion is largely one paragraph, many points should be expanded.
Author Response
Introduction
- Line 49: there’s a formatting issue here – the font changed
RESPONSE: Thank you for this remark. Changed.
- Paragraph that beings on line 52: I would like to understand more about the socioeconomic and cultural factors influencing the centralized v. uncentralized. Is it the richer countries that are more likely to offer GAC in large academic medical institutions? What is the history of this discrepancy? As we consider the implications for folks from the global south, these sociopolitical factors are important. “Common in European countries” has a lot under the surface
RESPONSE: Thank you for this remark. As it would be beyond the scope of the present paper, we were not able to address all the aspects you raised above. However, we discussed the issue in another paper. We referred to the issues and the publication as follows:
“Historically, transgender healthcare delivery has evolved differently in Europe and North America, whereas in other countries, especially those of the global south, structures are not even sufficiently developed to ensure a certain degree of access to specific care for TGD people11. A more detailed discussion on the issue can be found in another publication by the authors of the present study10.”
- The introduction would benefit from improved transitions between sections
RESPONSE: Thank you for this remark. We tried to smoothen the transitions between the sections.
- This is a good introduction to client centered care, more information about how transgender patients have experienced care historically and contextually would underscore the need for this approach
RESPONSE: Thank you for this remark. Again, we referred to a paper by our research group that discusses the history of delivery of transgender healthcare in detail. However, we also added the following section to point out why the issue of centralized and decentralized delivery of transgender healthcare might be important to investigate:
“Prior research on the influence of centralized and decentralized delivery of care has mostly been conducted investigating the issue from the perspective of healthcare policy, e.g., regarding financial benefits of decentralizing care7. In daily medical practice, decentralized delivery of care has been found to be suitable for certain condition (e.g., HIV), as it ensures easy access to care8,9. However, regarding transgender healthcare, centralized and decentralized delivery of has not been further investigated yet. As the setting of transgender healthcare delivery differs significantly between and within different countries, it could be considered a potential influence on the outcome and quality of care10.“
- Please provide more context for why each measure was chosen
RESPONSE: Thank you for this remark. We added the following section:
“All measures were chosen based on systematic reviews of the psychometric qualities of the questionnaires and the clinical experience of the research group.”
Method
- Please speak to the impact of having only single surgeon’s data set
RESPONSE: Thank you for this remark. We added the following section:
“Multi-center studies might be necessary to deal with potential effects by the clinics individual surgeon(s). All the participants of the current study were undergoing vaginoplasty by a single surgeon. Therefore, potential selection effects (e.g., specific patient population) and confounding variables (e.g., surgeon´s experience, surgical approach) cannot be ruled out.“
- Please speak to the impact of having a vaginoplasty at 16 v 40 developmentally.
Thank you for this remark. We added the following section:
“Additionally, there are also different experiences and considerations for individuals undergoing vaginoplasty depending on their age (e.g., for how long someone experienced gender dysphoria, surgical considerations), that might have influenced the outcomes of the present study. However, another analysis of the data focusing on the surgical outcome did not find the age to be different between participants accessing vaginoplasty in centralized and decentralized settings12.”
- The statistical analysis of this paper is beyond my scope of knowledge to opine on specifically.
Results
- Given that the sample is 85% German, it may behoove the generalization of the paper to focus on the population at hand.
RESPONSE: Thank you for this remark. We addressed the issue in the limitations section as follows:
“As the data were generated in Germany, a country that is ethnically rather homogenous, higher racial and ethnical diversity could also alter the results significantly. Moreover, Germanys system of mandatory health insurance founded by general wage contributions might influence the data compared to countries like the United states with a mix of public and private for-profit and non-profit insurers. „
- Can you please say more about the separation between woman/female and trans woman?
RESPONSE: We assume you are referring to the options participants had when giving information on their gender identity in the section on sociodemographic data. It has been found in prior research that gender identity is not solely determined by biological characteristics, but by a person’s individual gender-related experience. Therefore, we as a research group see the problems with the distinction made using the terms “woman” and “trans woman”. However, some transgender and gender-diverse people use the term “trans woman” rather than “woman” in a process of empowerment and self-affirmation. Therefore, we wanted to give participants the option to choose between both “woman” and “trans woman“ in the sociodemographic section.
Discussion
- Decentralized settings seem to offer a lot, though as the discussion points out, wrap around care may offer more in terms of psychological and general health support. Is higher personal responsibility positive?
RESPONSE: Thank you for this remark. It appears unclear to us to which specific aspect of the discussion you are referring to with that comment. However, we know that transgender and gender-diverse people often take high responsibility for their healthcare, educate themselves, and are well aware of the various effects gender-affirming procedures could have. Therefore, we understand high personal responsibility as positive.
- Post surgical problems are a huge factor in future satisfaction of care – can you please speak more to this?
RESPONSE: Thank you for this remark. We addressed the issue in the limitations section as follows:
“In addition, post-surgical complications might have influenced the assessment of patient-centeredness, too. A recent analysis by authors of the present study found that even though there is a considerable rate of smaller short-term post-surgical complications (41%), the number of severe complications is low (2%13). Systematic reviews of current evidence found comparable number14,15. Additionally, a recent systematic review found that a majority of over 90% of TGD people undergoing vaginoplasty report high satisfaction with the procedure and a regret rate of only 2%14. However, it is unclear if the high satisfaction with the procedure can be directly transferred to a high satisfaction with the various aspects of healthcare quality (e.g., client-centeredness).”
- The discussion is largely one paragraph, many points should be expanded.
RESPONSE: Thank you for this remark. As the present study is only a first exploratory approach to investigate centralized and decentralized delivery of transgender healthcare services, we want to be careful with overinterpreting the results. However, we added several additional aspects, especially to the limitations section, to make clear that our results might be influence by several factors and future research on the issue is crucially needed.
Round 2
Reviewer 2 Report
Reviewers have addressed comments adequately
Author Response
Thank you! We made some minor changes as suggested.